# Cardiovascular risk prediction using physical performance measures in COPD: results from a multicentre observational study

Jilles M Fermont [1,2] Marie Fisk [1] Charlotte E Bolton [3] William MacNee,[4] John R Cockcroft,[5] Jonathan Fuld,[6] Joseph Cheriyan [1] Divya Mohan,[7] Kaisa M Mäki-Petäjä [1] Ali B Al-Hadithi [1] Ruth Tal-Singer [7] Hana Müllerova,[8] Michael I Polkey [9] Angela M Wood [2,10,11,12,13] Carmel M McEniery,[1] Ian B Wilkinson,[1,14] on behalf of the ERICA consortium

For numbered affiliations see end of article.

**Correspondence to**
Dr Jilles M Fermont;
jmf88@medschl.cam.ac.uk

## ABSTRACT

**Objectives** Although cardiovascular disease (CVD) is a common comorbidity associated with chronic obstructive pulmonary disease (COPD), it is unknown how to improve prediction of cardiovascular (CV) risk in individuals with COPD. Traditional CV risk scores have been tested in different populations but not uniquely in COPD. The potential of alternative markers to improve CV risk prediction in individuals with COPD is unknown. We aimed to determine the predictive value of conventional CVD risk factors in COPD and to determine if additional markers improve prediction beyond conventional factors.

**Design** Data from the Evaluation of the Role of Inflammation in Chronic Airways disease cohort, which enrolled 729 individuals with Global Initiative for Chronic Obstructive Lung Disease (GOLD) stage II–IV COPD were used. Linked hospital episode statistics and survival data were prospectively collected for a median 4.6 years of follow-up.

**Setting** Five UK centres interested in COPD.

**Participants** Population-based sample including 714 individuals with spirometry-defined COPD, smoked at least 10 pack years and who were clinically stable for >4 weeks.

**Interventions** Baseline measurements included aortic pulse wave velocity (aPWV), carotid intima–media thickness (CIMT), C reactive protein (CRP), fibrinogen, spirometry and Body mass index, airflow Obstruction, Dyspnoea and Exercise capacity (BODE) Index, 6 min walk test (6MWT) and 4 m gait speed (4MGS) test.

**Primary and secondary outcome measures** New occurrence (first event) of fatal or non-fatal hospitalised CVD, and all-cause and cause-specific mortality.

**Results** Out of 714 participants, 192 (27%) had CV hospitalisation and 6 died due to CVD. The overall CV risk model C-statistic was 0.689 (95% CI 0.688 to 0.691). aPWV and CIMT neither had an association with study outcome nor improved model prediction. CRP, fibrinogen, GOLD stage, BODE Index, 4MGS and 6MWT were associated with the outcome, independently of conventional risk factors (p<0.05 for all). However, only 6MWT improved model discrimination (C=0.727, 95% CI 0.726 to 0.728).

### Strengths and limitations of this study

► This is the first study assessing the utility of conventional cardiovascular (CV) disease risk factors for CV risk prediction and the value of additional markers of risk within a chronic obstructive pulmonary disease (COPD) cohort.

► Patient-level cohort data were linked to hospital admission data (ie, hospital episode statistics) obtained from the National Health Services in England, Scotland and Wales, and Office for National Statistics record of mortality with analyses limited to 5 years of follow-up.

► Hospitalised CV episodes were coded based on International Statistical Classification of Diseases and Related Health Problems, 10th Revision, classifications extracted from both primary and secondary positions.

► A multivariable prediction model, with 10-fold cross validation and 200 replications, was used to evaluate a wide range of CV and physical performance biomarkers.

► Generalisability of our results is limited to those with moderate COPD in the UK with NHS hospitalisations.

**Conclusion** Poor physical performance defined by the 6MWT improves prediction of CV hospitalisation in individuals with COPD.

**Trial registration number** ID 11101.

## INTRODUCTION

Chronic obstructive pulmonary disease (COPD) is associated with a twofold to three-fold increased risk of coronary artery disease and other cardiovascular (CV) comorbidities.[1] [2] COPD and cardiovascular disease (CVD) are common conditions that share important common risk factors, such as smoking and physical inactivity, and both tend to affect older people. In addition,

both conditions are often associated with raised systemic inflammatory markers and increased arterial stiffness.[3][4]

Given the global healthcare burden associated with CVD,[5] there is incentive to improve the accuracy of CV risk prediction in different populations. Individuals with COPD may be considered constitutively at high CV risk, given their age and smoking history. However, it is unknown whether classic CV risk prediction models, such as the Framingham General CV Risk score, which predicts an individual's 10-year risk of developing CVD based on an algorithm of weighted risk factors and has been tested in a number of different populations,[6–9] perform well in individuals with COPD.[10][11]

Conceptually, the performance of the Framingham model might be improved by additional measures. Candidate measurements include surrogate CV risk markers that impart mechanistic information (ie, aortic pulse wave velocity (aPWV) or carotid intima–media thickness (CIMT)), inflammatory markers (ie, C reactive protein (CRP)) or measures of physical performance that enhance CV risk prediction in individuals with COPD. In fact, the value of these measures in CV risk prediction have been explored in different population groups. For example, the 6 min walk test (6MWT) significantly improved risk prediction in patients with stable coronary disease.[12] In individuals with intermediate CV risk but without CVD, adding CRP or fibrinogen to conventional risk factors modestly improved CV event prediction.[13] A meta-analysis of aPWV studied in different disease cohorts showed it improved CV event prediction independently of conventional risk factors.[14] In contrast, CIMT measurement, although associated with CV risk factors, did not significantly improve risk prediction beyond traditional factors in individuals with hypertension.[15] Given that individuals with COPD, in addition to having high CV risk based on conventional factors, also have increased aPWV, CIMT[4] and increased inflammatory markers,[16] as well as reduced lung function, which is associated with increased CV risk in the general population[17] and reduced physical performance,[18][19] the clinical significance of these findings in relation to CV risk prediction in individuals with COPD is an important question to address.

The aims of our study were to first determine the predictive value of conventional CVD risk factors for CV risk, defined by new occurrence (first event since study enrolment) of fatal or non-fatal hospitalised CVD in individuals with stable Global Initiative for Chronic Obstructive Lung Disease (GOLD) stage II–IV[20] COPD. A secondary aim was to determine whether addition of alternative CV measures (ie, aPWV and CIMT), inflammatory markers (ie, CRP and fibrinogen), COPD severity (ie, GOLD stage and Body mass index (BMI), airflow Obstruction, Dyspnoea and Exercise capacity (BODE) Index), as well as physical performance tests commonly used in COPD research (ie, 6MWT and 4 m gait speed (4MGS)) improved the predictive value of a CV risk model based on conventional CVD risk factors for CV risk prediction. We addressed these questions in the Evaluation of the Role of Inflammation in Chronic Airways disease (ERICA) COPD cohort using study data and linked UK electronic health records.

## METHODS
### Study design and participants
The ERICA study is a multicentre, observational cohort study with 729 individuals with stable GOLD stage II–IV[20] COPD, established to identify important CV and physical performance biomarkers that could be targeted to improve the outcomes of individuals with COPD. Participants had a clinical diagnosis of COPD, smoking history of at least 10 pack years, postbronchodilator forced expiratory lung volume in one second ($FEV_1$)/forced vital capacity ratio of <0.7 and $FEV_1$ ≤80% of predicted normal lung function, and were aged >40 years old and clinically stable for >4 weeks. Full details of the protocol have been provided elsewhere.[21] Baseline data captured included demographics, spirometry, blood circulating biochemical markers, measures of arterial stiffness (ie, aPWV and Augmentation Index (AIx)), CIMT and physical performance (ie, 4MGS and 6MWT). Individuals in the ERICA study were linked with UK National Health Services (NHS) electronic healthcare records (ie, hospital episode statistics (HES) data are a database that includes details of all hospital admissions, accident and emergency department visits and outpatient appointments at an individual patient level)[22] and Office for National Statistics death data through anonymised identifications provided by the NHS.

### Clinical measures
After 4 hours of fasting, with no bronchodilators for 6 hours, and 10 minutes of supine rest, carotid–femoral aPWV and AIx measurements were taken using a SphygmoCor system as previously described.[23] CIMT of the common carotid arteries was measured using B-mode ultrasound at a distance of 1 cm from the carotid bulb with a linear probe of 7–12 MHz.[24] The thickest artery of the two was included in the analysis. Fasting blood samples were taken for biochemical analysis, including plasma fibrinogen, serum CRP and glucose. Physical performance measures 6MWT[25] and 4MGS[26] were assessed according to guidelines. Diabetes status and antihypertensive treatment were self-reported at baseline study visit. Disease severity was defined according to GOLD classification.[20] Points for the BODE Index were assigned as described by Celli *et al*.[27]

### CV hospitalisation and mortality
CV hospitalisation and mortality data were extracted from the linked hospital admission data and death certificates. Non-fatal CV episodes were extracted from both primary and secondary International Statistical Classification of Diseases and Related Health Problems, 10th Revision (ICD-10) coding positions. Causes of death were adjudicated by CV and pulmonary physicians. We defined

the primary outcome as first reported occurrence (since study enrolment) of fatal or non-fatal hospitalised CVD, where CVD was defined as disease of the arteries, stroke or heart failure (see online supplemental table 1) based on classifications used by the Emerging Risk Factors Collaboration.[28] Time to primary outcome was calculated from the difference between the baseline visit date (starting December 2011) and either the date of death or first hospitalised CV attendance up to November 2017, when follow-up discontinued. Secondary outcomes of interest were all-cause and cause-specific mortality (defined as CV, pulmonary, cancer or other).

### Risk factors of interest

Conventional CVD risk factors included age, sex, self-reported smoking status (current/ex-smoker), high-density lipoprotein (HDL) cholesterol, total cholesterol, systolic blood pressure (SBP), diabetes (yes/no) and treatment for high blood pressure (yes/no). We also assessed the addition of the following risk factors: BMI, aPWV and CIMT, fasting glucose, CRP and fibrinogen, COPD severity (ie, GOLD stage and BODE Index) and measures of physical performance (ie, 6MWT and 4MGS).

### Statistical analysis

HRs with 95% CIs were estimated using Cox regression models stratified by study centre. Age and sex were added to all models. To quantify the independent association of CIMT, we further included SBP. For aPWV, we also included mean arterial pressure and resting heart rate. For AIx, we additionally included resting heart rate and height. Proportional hazards were assessed by Schoenfeld's global tests. We assessed the relationships of the new markers and outcomes and consequently log-transformed the following risk factors: CRP, fibrinogen and glucose. HRs for log-transformed risk factors represented a twofold increase in the risk factor, whereas others were presented as a change in unit. We evaluated the predictive value of new markers added to the conventional CVD risk factors using measures of discrimination (ie, Harrell's C-statistic)[29 30] and calibration (ie, Gronnesby and Borgan test and Brier score). The Gronnesby and Borgan test is an overall calibration test for Cox models based on grouping individuals by their estimated risk score and compares observed and model-based expected events within each group (a p value of >α suggests no difference). Brier scores range from 0 to 1 (0 is perfect accuracy and 1 is perfect inaccuracy) and allow comparison of performance of a model with a reference model. The C-statistic is a measure for validating the discriminative ability of a model. Values range from 0.5 to 1.0 (1.0 is a perfect prediction and 0.5 is a random guess). A higher score indicates better discriminative ability of the model. To optimise efficiency and to avoid optimism from internal validation in small samples, we used 10-fold cross validation with 200 replications[31] (see online supplemental text 1 and online supplemental figures 1–5 for further statistical analyses details).

| Table 1 | Baseline characteristics (N=714) |
|---|---|
| **Characteristics** | **Summary measures** |
| Conventional CVD risk factors | |
| Age (years) | 67 (62–73) |
| Male | 434 (61) |
| Current smoker | 218 (31) |
| HDL (mmol/L) | 1.4 (1.2–1.7) |
| Cholesterol (mmol/L) | 5.0 (4.3–5.8) |
| SBP (mm Hg) | 142 (131–154) |
| Diabetes mellitus | 82 (12) |
| Drugs to treat hypertension | 245 (34) |
| CV measures | |
| aPWV (m/s) | 9.8 (8.4–11.8) |
| CIMT (mm) | 0.81 (0.71–0.96) |
| Alternative measures | |
| CRP (mg/L) | 1.21 (0.47–2.01) |
| Fibrinogen (g/dL) | 1.22 (1.06–1.36) |
| Glucose (mmol/L) | 1.59 (1.46–4.59) |
| BMI (kg/m$^2$) | 27 (23–31) |
| GOLD (stage) | 2 (2–3) |
| 4MGS (m/s) | 0.95 (0.77–1.14) |
| 6MWT distance (m) | 366 (255–440) |
| BODE (point) | 3 (1–5) |

Values are given as median and IQR, or number of cases (%). aPWV, aortic pulse wave velocity; BMI, body mass index; BODE, Body mass index, airflow Obstruction, Dyspnoea and Exercise capacity; CIMT, carotid intima–media thickness; CRP, C reactive protein; CV, cardiovascular; CVD, cardiovascular disease; GOLD, Global Initiative for Chronic Obstructive Lung Disease; HDL, high-density lipoprotein; 4MGS, 4 m gait speed; 6MWT, 6 min walk test; SBP, systolic blood pressure.

Observational data are reported according to the Strengthening the Reporting of Observational Studies in Epidemiology statement.[32] All tests were two-sided and statistical significance was defined by 95% CI for HRs not traversing 1 or p<0.05. Our analyses were performed using STATA V.13 and R (R Foundation).

### RESULTS

Of the 729 individuals included in the study, 714 (98%) could be linked with hospital admission and survival records, and were included in the analysis (online supplemental figure 6). The median age was 67 (IQR 62–73) years, and 434 (61%) individuals were male (table 1). A third (n=218) of the cohort smoked; 12% (n=82) had self-reported diabetes; 34% (n=245) were taking antihypertensive medications; and 31% (n=224) were taking cholesterol-lowering medications at baseline. The median FEV$_1$ was 1.3 L (0.9–1.7 L) (mean±SD=1.34±0.53). In total, 192 individuals (27%) had a first event of CV hospitalisation (peripheral arterial disease (n=9), diseases of

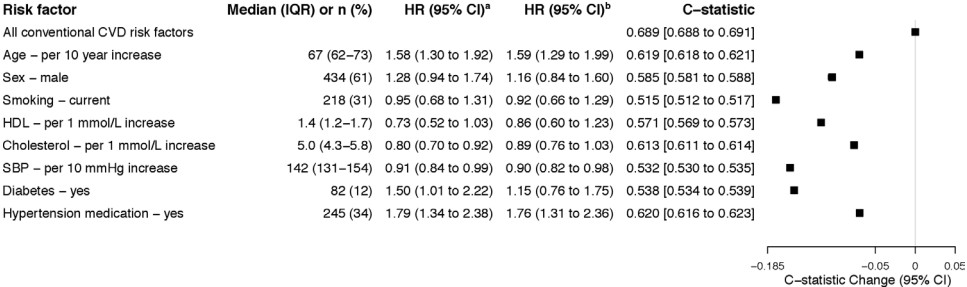

| Risk factor | Median (IQR) or n (%) | HR (95% CI)a | HR (95% CI)b | C–statistic |
|---|---|---|---|---|
| All conventional CVD risk factors | | | | 0.689 [0.688 to 0.691] |
| Age – per 10 year increase | 67 (62–73) | 1.58 (1.30 to 1.92) | 1.59 (1.29 to 1.99) | 0.619 [0.618 to 0.621] |
| Sex – male | 434 (61) | 1.28 (0.94 to 1.74) | 1.16 (0.84 to 1.60) | 0.585 [0.581 to 0.588] |
| Smoking – current | 218 (31) | 0.95 (0.68 to 1.31) | 0.92 (0.66 to 1.29) | 0.515 [0.512 to 0.517] |
| HDL – per 1 mmol/L increase | 1.4 (1.2–1.7) | 0.73 (0.52 to 1.03) | 0.86 (0.60 to 1.23) | 0.571 [0.569 to 0.573] |
| Cholesterol – per 1 mmol/L increase | 5.0 (4.3–5.8) | 0.80 (0.70 to 0.92) | 0.89 (0.76 to 1.03) | 0.613 [0.611 to 0.614] |
| SBP – per 10 mmHg increase | 142 (131–154) | 0.91 (0.84 to 0.99) | 0.90 (0.82 to 0.98) | 0.532 [0.530 to 0.535] |
| Diabetes – yes | 82 (12) | 1.50 (1.01 to 2.22) | 1.15 (0.76 to 1.75) | 0.538 [0.534 to 0.539] |
| Hypertension medication – yes | 245 (34) | 1.79 (1.34 to 2.38) | 1.76 (1.31 to 2.36) | 0.620 [0.616 to 0.623] |

**Figure 1** Conventional CVD risk factors at baseline, their HRs and discriminative ability for fatal or non-fatal hospitalised CVD. Values are given as median and IQR, or number of cases (%). Baseline data of 714 patients are included. All models are stratified by recruitment site. There were <5% missing values for descriptive variables such as body mass index and smoking status. Missing values were addressed using multiple imputations using chained equations. aModel includes age and sex. bModel includes conventional CVD risk factors: age, sex, smoking, HDL, total cholesterol, SBP, diabetes and hypertension medication. CVD, cardiovascular disease; HDL, high-density lipoprotein; SBP, systolic blood pressure.

arteries, arterioles and capillaries (n=7), angina (n=21), unstable angina (n=3), coronary heart disease not otherwise specified (n=63), acute myocardial infarction (MI) and certain current complications following acute MI (n=11), cerebral infarction (n=11), stroke, not specified as haemorrhage or infarction (n=3), other stroke (n=18), heart failure (n=32) and abdominal aortic aneurysm (n=8); n=116 (60%) were in ICD-10 secondary coding position) during median follow-up for 4.6 years, and 6 individuals had CV death without any preceding CV episode. CV hospitalisation accounted for the majority (97%) of events analysed. The CV incidence rate was 6.7 (95% CI 5.8 to 7.7) per 100 person-years (see online supplemental tables 1 and 2) for categorisation of different admission codes for CV hospitalisation.

### Conventional CVD risk factors

Of the conventional CVD risk factors, age and treatment for high blood pressure had significant positive associations with the study's outcome, whereas SBP had a significant negative association and other risk factors (ie, sex, smoking status, cholesterol, HDL cholesterol and diabetes) were not significantly associated with CV hospitalisation. Use of hypertension drug treatment followed by age and total cholesterol contributed most to the discriminative ability of the model. The overall discriminative ability of the CV risk model had a C-statistic of 0.689 (95% CI 0.688 to 0.691, figure 1 and online supplemental table 3).

### Surrogate CV risk markers

Except for AIx, neither aPWV nor CIMT was significantly associated with CV hospitalisation after including conventional CVD risk factors (figure 2 and online supplemental table 4). Moreover, none of the CV risk markers significantly changed the discriminative ability of the CV risk model.

### Physical performance measures, COPD severity and inflammatory markers

Multivariable analysis identified that poor physical performance (ie, reduced 6MWT distance and slower 4MGS) was significantly associated with increased CV hospitalisation, independently of conventional CVD risk factors. With the exception of glucose and BMI, severity of COPD defined by higher BODE Index and GOLD stage, as well

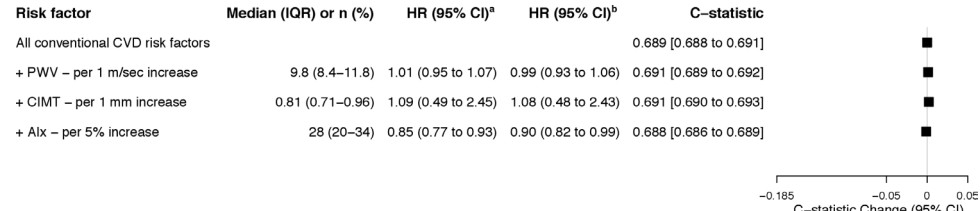

| Risk factor | Median (IQR) or n (%) | HR (95% CI)a | HR (95% CI)b | C–statistic |
|---|---|---|---|---|
| All conventional CVD risk factors | | | | 0.689 [0.688 to 0.691] |
| + PWV – per 1 m/sec increase | 9.8 (8.4–11.8) | 1.01 (0.95 to 1.07) | 0.99 (0.93 to 1.06) | 0.691 [0.689 to 0.692] |
| + CIMT – per 1 mm increase | 0.81 (0.71–0.96) | 1.09 (0.49 to 2.45) | 1.08 (0.48 to 2.43) | 0.691 [0.690 to 0.693] |
| + AIx – per 5% increase | 28 (20–34) | 0.85 (0.77 to 0.93) | 0.90 (0.82 to 0.99) | 0.688 [0.686 to 0.689] |

**Figure 2** Aortic stiffness at baseline, their HRs and discriminative ability for fatal or non-fatal hospitalised CVD. Values are given as median and IQR, or number of cases (%). Baseline data of 714 patients are included. All models are stratified by recruitment site. Gronnesby and Borgan goodness of fit ($\chi^2$(3), p>$\chi^2$): CV risk model (2.07, 0.559), aPWV (1.64, 0.652), CIMT (2.32, 0.509), and AIx (3.08, 0.380). Estimates based on quartiles of risk. Brier score: CV risk model 0.129 (95% CI 0.111 to 0.146), aPWV 0.126 (95% CI 0.108 to 0.145), CIMT 0.128 (95% CI 0.110 to 0.147) and AIx 0.126 (95% CI 0.109 to 0.144). Lower score indicates better accuracy of estimates. aModel includes age and sex. bModel includes conventional CVD risk factors: age, sex, smoking, high-density lipoprotein, total cholesterol, SBP, diabetes and hypertension medication. CIMT further included SBP. Carotid–femoral aPWV further included mean arterial pressure and resting heart rate. AIx further included resting heart rate and height. There were about 10% missing values for variables CIMT (n=66) and aPWV (n=60). Missing values were addressed using multiple imputations using chained equations. AIx, Augmentation Index; aPWV, aortic pulse wave velocity; CIMT, carotid intima–media thickness; CV, cardiovascular; CVD, cardiovascular disease; PWV, pulse wave velocity; SBP, systolic blood pressure.

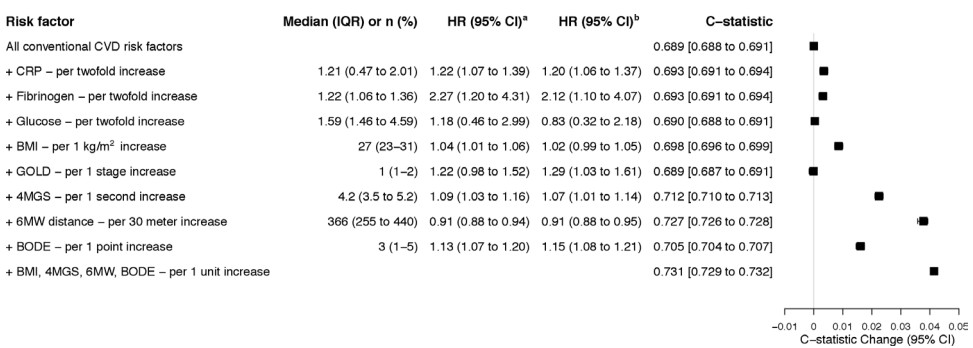

| Risk factor | Median (IQR) or n (%) | HR (95% CI)[a] | HR (95% CI)[b] | C–statistic |
|---|---|---|---|---|
| All conventional CVD risk factors | | | | 0.689 [0.688 to 0.691] |
| + CRP – per twofold increase | 1.21 (0.47 to 2.01) | 1.22 (1.07 to 1.39) | 1.20 (1.06 to 1.37) | 0.693 [0.691 to 0.694] |
| + Fibrinogen – per twofold increase | 1.22 (1.06 to 1.36) | 2.27 (1.20 to 4.31) | 2.12 (1.10 to 4.07) | 0.693 [0.691 to 0.694] |
| + Glucose – per twofold increase | 1.59 (1.46 to 4.59) | 1.18 (0.46 to 2.99) | 0.83 (0.32 to 2.18) | 0.690 [0.688 to 0.691] |
| + BMI – per 1 kg/m² increase | 27 (23–31) | 1.04 (1.01 to 1.06) | 1.02 (0.99 to 1.05) | 0.698 [0.696 to 0.699] |
| + GOLD – per 1 stage increase | 1 (1–2) | 1.22 (0.98 to 1.52) | 1.29 (1.03 to 1.61) | 0.689 [0.687 to 0.691] |
| + 4MGS – per 1 second increase | 4.2 (3.5 to 5.2) | 1.09 (1.03 to 1.16) | 1.07 (1.01 to 1.14) | 0.712 [0.710 to 0.713] |
| + 6MW distance – per 30 meter increase | 366 (255 to 440) | 0.91 (0.88 to 0.94) | 0.91 (0.88 to 0.95) | 0.727 [0.726 to 0.728] |
| + BODE – per 1 point increase | 3 (1–5) | 1.13 (1.07 to 1.20) | 1.15 (1.08 to 1.21) | 0.705 [0.704 to 0.707] |
| + BMI, 4MGS, 6MW, BODE – per 1 unit increase | | | | 0.731 [0.729 to 0.732] |

−0.01 0 0.01 0.02 0.03 0.04 0.05
C–statistic Change (95% CI)

**Figure 3** Alternative measures at baseline, their HRs and discriminative ability for fatal or non-fatal hospitalised CVD. Values are given as median and IQR, or number of cases (%). Baseline data of 714 patients are included. All models are stratified by recruitment site. Gronnesby and Borgan goodness of fit ($\chi^2$(3), p>$\chi^2$): CV risk model (2.07, 0.559); CRP (0.32, 0.956); fibrinogen (1.48, 0.687); glucose (0.42, 0.936); BMI (1.56, 0.668); GOLD (5.63, 0.131); 4MGS (4.70, 0.195); 6MWT (2.94, 0.401); BODE (6.46, 0.091); BMI, 4MGS, 6MWT, bode (4.12, 0.249). Estimates based on quartiles of risk. Brier score: CV risk model 0.129 (95% CI 0.111 to 0.146); CRP 0.125 (95% CI 0.107 to 0.142); fibrinogen 0.128 (95% CI 0.111 to 0.146); glucose 0.128 (95% CI 0.111 to 0.146); BMI 0.128 (95% CI 0.111 to 0.146); GOLD 0.128 (95% CI 0.110 to 0.146); 4MGS 0.127 (95% CI 0.110 to 0.144); 6MWT 0.123 (95% CI 0.105 to 0.140); BODE 0.124 (95% CI 0.106 to 0.142); BMI, 4MGS, 6MWT, BODE 0.122 (95% CI 0.104 to 0.140). Lower score indicates better accuracy of estimates. [a]Model includes age and sex. [b]Model includes conventional CVD risk factors: age, sex, smoking, high-density lipoprotein, total cholesterol, systolic blood pressure, diabetes and hypertension medication. There were <5% missing values for biochemical markers, including fibrinogen and cholesterol. Missing values were addressed using multiple imputations using chained equations. 4MGS, 4 m gait speed; 6MWT, 6 min walk test; BMI, body mass index; BODE, Body mass index, airflow Obstruction, Dyspnoea, and Exercise capacity; CRP, C reactive protein; CV, cardiovascular; CVD, cardiovascular disease; GOLD, Global Initiative for Chronic Obstructive Lung Disease.

as inflammatory markers (ie, CRP and fibrinogen), all had significant positive associations with CV hospitalisations independently of conventional CVD risk factors.

Predictive modelling indicated significant improvement in risk discrimination when adding BMI (C=0.698, 95% CI 0.696 to 0.699), BODE (C=0.705, 95% CI 0.704 to 0.707), 4MGS (C=0.712, 95% CI 0.710 to 0.713) or 6MWT (C=0.727, 95% CI 0.726 to 0.728) to the CV risk model, but not GOLD stage, inflammatory markers or glucose (figure 3 and online supplemental table 5). Calibration tests indicate good model fit (figures 2 and 3). Adding BMI, 4MGS, 6MWT and BODE collectively to the CV risk model resulted in a C-statistic of 0.731 (95% CI 0.729 to 0.732), indicating 6MWT primarily accounted for improved discriminative ability of the model. The model including 6MWT had a better Brier score relative to the CV risk model (0.123 vs 0.129, respectively).

### All-cause and cause-specific mortality

There were 144 deaths in total (nearly 20% of the cohort) over a median follow-up period of 4.6 years. The majority of deaths were in men (n=96, 67%), and pulmonary disease was the leading cause in both sexes (table 2). Pulmonary disease accounted for 65% of deaths in women and 50% in men, followed by cancer (19% in women, 27% in men), CV (6% and 16%) and other (10% and 7% in women and men, respectively).

### DISCUSSION

This is the first study assessing the utility of conventional CVD risk factors for CV risk prediction and the value of additional markers of risk within a COPD cohort.

Novel findings include (1) poor physical performance improved the discriminative power when added to the CV risk model; (2) BODE Index, GOLD stage, 4MGS and systemic inflammatory markers were positively associated with CV hospitalisations independently of conventional CVD risk factors, although they collectively improved the model's discriminative ability marginally; and (3) of the conventional CVD risk factors, age, SBP and use of antihypertensives were positively associated with study outcome. However, age, cholesterol and use of antihypertensives contributed most to the model's prognostic power. SBP had a significant negative association with CV hospitalisations but did not add to the model's predictive ability. Furthermore, aPWV and CIMT, despite providing in vivo mechanistic information about the arterial system, had no significant association with CV hospitalisations, whereas AIx did. Therefore, these data suggest that in COPD, physical performance (assessed by 6MWT) contributes to CV risk estimation defined predominantly

**Table 2** Mortality in the Evaluation of the Role of Inflammation in Chronic Airways disease cohort

| Cause-specific mortality | Female, n (%) | Male, n (%) | Total, N (%) |
|---|---|---|---|
| Pulmonary | 31 (65) | 48 (50) | 79 (55) |
| Cardiovascular | 3 (6) | 15 (16) | 18 (13) |
| Cancer | 9 (19) | 26 (27) | 35 (24) |
| Other | 5 (10) | 7 (7) | 12 (8) |
| All-cause | 48 (100) | 96 (100) | 144 (100) |

Deaths recorded over a median follow-up of 4.6 years.

by non-fatal CV hospitalisations. Finally, CVD contributed to only a small number of deaths at a median follow-up of 4.6 years. Pulmonary disease, followed by cancer, was the major cause of death in both men and women with COPD.

We observed a C-statistic of 0.689 based on the conventional CV risk model, which increased to a maximum of 0.731 when adding additional markers; this increment was primarily due to 6MWT. These C-statistic values indicate a moderate predictive ability of the models for the study's outcome. For perspective, a C-statistic of 0.53 was observed in a Framingham model in the very elderly (aged 85-plus years; the median age of our cohort was 67 years) without prior CVD, indicating it does not predict CV mortality in this group.[6] However, in primary care individuals without CVD in the Framingham study, a C-statistic of 0.76 for an outcome of general CVD (ie, coronary heart disease, stroke, peripheral artery disease or heart failure) demonstrated good discrimination.[10] Since our cohort included individuals with CVD at baseline, and different CVD risk factors and CV outcomes are evaluated in various studies, no direct comparison of study results can be made. That aPWV was not predictive in our cohort contrasts with findings reported by Ben-Shlomo *et al.* However, from the 17 included cohorts in their study, none were COPD cohorts; the mean age of the cohorts was lower; and the proportion of individuals taking drugs to treat hypertension was higher.[14]

Overall, our study emphasises the importance of physical function as a predictor of CV risk. Tests such as 6MWT or 4MGS are proxy measures of overall mobility and physical functioning.[33] Exercise capacity and CV fitness are known to be associated with fatal and non-fatal CVD,[34] while exercise-based cardiac rehabilitation reduces risk of CV events.[35] The 6MWT distance is prognostic in patients with stable coronary heart disease[12] and in those with moderate-to-severe heart failure.[36]

In our cohort, poor physical function improved discriminative ability of the CV risk model beyond conventional CVD factors. This has implications for clinical practice in CV risk assessment for individuals with COPD, suggesting it may be helpful to incorporate physical performance into clinical assessment. Especially those aged under 65 years may benefit most from active CVD assessment, according to Morgan *et al.*[37] However, given that 6MWT can be logistically challenging to set up and time-consuming,[38] the faster and simpler 4MGS that also significantly improved the C-statistic could be used as a simpler alternative. In elderly with CVD, 4MGS is comparable to 6MWT in predicting all-cause mortality.[39]

The BODE Index also significantly improved the C-statistic of the CV risk model. However, this was primarily due to the 6MWT component. In fact, although GOLD stage had a significant positive association with the study outcome, this did not improve the prognostic power of the model. Therefore, despite the association between airflow limitation and CV risk in general population studies,[17 40] in patients with COPD, physical performance

assessment rather than another component of BODE (ie, spirometry) adds value to CV risk prediction.

Although we did not find an association between inflammatory markers and surrogate markers of CV risk in the baseline cross-sectional component of the ERICA study,[41] both CRP and fibrinogen were associated with CV hospitalisations captured over nearly 5 years of follow-up. That the associations remained significant after including conventional CVD risk factors indicates potential value for identifying high-risk individuals within a COPD population. The inverse relationship of SBP was an unexpected finding. One hypothesis is that SBP has a J-shaped curve for CV events and mortality.[42] Therefore, in this cohort, lower SBP might be a marker of sickness and frailty, hence its association with CV hospitalisations.

A third of the cohort had a CV-related hospitalisation during follow-up, and the majority had pre-existing CV comorbidity at baseline. The high numbers of CV hospitalisations give perspective of what this comorbidity incurs for patients and the huge healthcare costs involved. In contrast to the sizeable number of CV hospitalisations was the small number of CV deaths. Gayle *et al* previously reported that CV-related mortality in patients with chronic lung disease had already started to decline in England.[43] This may reflect better CV risk management reducing CV mortality. Importantly though, such risk management does not seem to impact CV morbidity and is an area that requires future research to determine the optimum approach to impact CV morbidity in individuals with COPD.

Nearly 20% of the cohort died during follow-up. This is comparable to the TOwards a Revolution in COPD Health (TORCH) study where approximately 15% of the cohort died over 3 years of follow-up.[2] Pulmonary disease, followed by cancer, accounted for proportionally more deaths in our cohort compared with TORCH, whereas CV-specific mortality was less (10%–15% vs 27% in TORCH). Reasons for these findings are not entirely understood, but the sizeable proportion of our cohort already on medicines for dyslipidaemia and blood pressure control may be an important factor.

Our study has limitations. The conventional CVD risk factors are usually used to predict CV risk defined by development of CVD. However, the majority of our participants already had CVD and were on CV medications, which may be a confounding factor impacting the discriminative ability of CVD risk factors. Moreover, in our study, CV risk was defined differently (as CV hospitalisation with CV mortality). Hospitalised CV episodes were coded based on ICD-10 classifications[28] extracted from both primary and secondary positions. Notably, most CV hospitalisations were recorded in secondary positions, indicating that the primary admission might be related to something else. Due to the few CV events recorded in the primary position, we were unable to conduct a sensitivity analysis including CV events in the primary position only. The study period covered the time from study enrolment until the end of study or death. Some

individuals, however, may have been admitted to hospital for CV events before study enrolment. We were unable to obtain HES data outside this period. Competing risks may have occurred, for example, individuals who died of other causes may not have experienced a CV event during the study period for this reason. However, our study did not have sufficient statistical power to assess this. It is possible our models predict non-CV hospitalisation due to, for example, potential cross-contamination of COPD and CV hospitalisation as a result of misclassification of morbidity and mortality that often occurs when data are obtained from routine sources.[44] Deprivation scores may be another important determinant of CV risk in individuals with COPD that is worth evaluating in future research,[45] although the relatively small cohort size of our study meant we did not include this in the analysis.

Furthermore, follow-up time for the study was a maximum of 5 years, whereas risk scores such as Framingham Risk Score and QRISK calculate a 10-year risk. Hence, the prognostic value of variables may alter with a different time horizon and depends on the extent of time trends in the new biomarkers. For example, a too short time period may result in an insufficient number of events, while over a longer time period, for example, 20 years, the predictive ability would diminish because ageing is a strong predictor. We did not have access to an independent validation cohort but used cross-validation techniques instead. Generalisability of our results is limited to those with moderate COPD in the UK with NHS hospitalisations.

## CONCLUSION

In a UK COPD cohort, poor physical performance assessed by 6MWT or 4MGS and inflammatory biomarkers are associated with subsequent CV-related hospitalisations, independently of conventional CVD risk factors. Importantly, poor physical performance (defined primarily by 6MWT) also significantly improved the predictive discrimination of the CV risk model. These data suggest an assessment of physical performance may enhance CV risk evaluation in individuals with COPD.

**Author affiliations**
[1]Division of Experimental Medicine and Immunotherapeutics, Department of Medicine, University of Cambridge, Cambridge, UK
[2]British Heart Foundation Cardiovascular Epidemiology Unit, Department of Public Health and Primary Care, University of Cambridge, Cambridge, UK
[3]Division of Respiratory Medicine and NIHR Nottingham BRC respiratory theme, University of Nottingham, Nottingham, UK
[4]Centre for Inflammation Research, Queen's Medical Research Institute, The University of Edinburgh, Edinburgh, UK
[5]Department of Cardiology, Columbia University Medical Center, New York City, New York, USA
[6]Department of Respiratory Medicine, Cambridge University Hospitals NHS Foundation Trust, Cambridge, UK
[7]Medical Innovation, Value Evidence Outcomes, GSK R&D, Philadelphia, Pennsylvania, USA
[8]GSK R&D, Uxbridge, UK
[9]Department of Respiratory Medicine, Royal Brompton Hospital, London, UK
[10]British Heart Foundation Centre of Research Excellence, University of Cambridge, Cambridge, UK
[11]National Institute for Health Research Blood and Transplant Research Unit in Donor Health and Genomics, University of Cambridge, Cambridge, UK
[12]National Institute for Health Research Cambridge Biomedical Research Centre, University of Cambridge and Cambridge University Hospitals, Cambridge, UK
[13]Health Data Research UK Cambridge, Wellcome Genome Campus and University of Cambridge, Cambridge, UK
[14]Cambridge Clinical Trials Unit, Cambridge University Hospitals NHS Foundation Trust, Addenbrooke's Hospital, Cambridge, UK

**Correction notice** This article has been corrected since it first published. The provenance and peer review statement has been included.

**Contributors** The following authors made substantial contributions to the conceptualisation or design: JMF, RT-S, HM, MIP, AMW and IBW; data curation: JMF, IBW, KM-P and ABA-H; formal analysis: JMF; interpretation: JMF, MF and IBW; funding acquisition: RT-S, MIP and IBW; investigation: JMF, MF, CEB, WMcN, JRC, JF, JC, DM, KM-P, ABA-H, MIP, CMMcE and IBW; methodology: JMF, AMW and IBW; project administration: JMF, RT-S, MIP and IBW; resources: JMF, MF, CEB, WMcN, JRC, DM, MIP, CMMcE and IBW; software: JMF; supervision: DM, RT-S, HM, MIP, AMW and IBW; validation: JMF; visualisation: JMF; and writing (original draft preparation): JMF. All authors contributed to the reviewing and editing of the manuscript.

**Funding** This work was supported by a grant (9157–61188) from Innovate UK (formerly known as Technology Strategy Board) with contributory funding in kind (eg, scientific expertise and meeting rooms) from GSK, a consortium partner, who also funded the corresponding author's PhD. As a consortium partner, GSK was involved in the study design, data analysis, decision to publish and preparation of the manuscript. The specific roles of all authors are articulated in the Author contributions section. RT-S was a coinvestigator on the grant and as a consortium member was involved in the decision to publish, and preparation of the manuscript. IBW, JC and CMMcE acknowledge funding support from theNational Institute for Health Research (NIHR) Cambridge Comprehensive Biomedical Research Centre. CEB is supported by the NIHR Nottingham BRC respiratory theme. This work was supported by core funding from the UK Medical Research Council (MR/L003120/1), the British Heart Foundation (RG/13/13/30194 and RG/18/13/33946) and the NIHR (Cambridge Biomedical Research Centre at the Cambridge University Hospitals NHS Foundation Trust). This work was also supported by Health Data Research UK, which is funded by the UK Medical Research Council, Engineering and Physical Sciences Research Council, Economic and Social Research Council, Department of Health and Social Care (England), Chief Scientist Office of the Scottish Government Health and Social Care Directorates, Health and Social Care Research and Development Division (Welsh Government), Public Health Agency (Northern Ireland), British Heart Foundation and Wellcome.

**Disclaimer** The views and opinions expressed are those of the authors and do not necessarily reflect those of the University of Cambridge, the NHS, the National Institute for Health Research, the Department of Health and Social Care or other parent institutions of the authors. The corresponding author and IBW had full access to all the data in the study and takes responsibility for the integrity of the data and the accuracy of the data analysis.

**Competing interests** GSK, a consortium partner, funded JMF's PhD. HM, RT-S and DM were employees of GSK at the time this work was completed and own GSK shares and stock options. JC is employed by Cambridge University Hospitals NHS Foundation Trust and is obligated to spend 50% of his time on GSK clinical trial activity, representing a significant relationship; however, he receives no other benefits or compensation from GSK. MIP received grants from GSK outside the submitted work. IBW received grants from GSK during the conduct of the study and outside the submitted work.

**Patient consent for publication** Not required.

**Ethics approval** Informed patient consent was obtained in writing from all study participants and permits the publishing of all data included in this article. The investigation conforms with the principles outlined in the Declaration of Helsinki. Ethical approval was granted by the National Research Ethics Service Committee East of England, Cambridge South (11/EE/0357).

**Provenance and peer review** Not commissioned; externally peer reviewed.

**Data availability statement** Data are available upon reasonable request. Data may be obtained from a third party and are not publicly available. The dataset

underlying these findings, with deidentified participant data (including the data dictionary), are available to interested and qualified researchers upon request and can be obtained from the Cambridge Clinical Trials Unit. Access to hospital episode statistics requires a data sharing agreement with the National Health Services. For data access, please contact cctu@addenbrookes.nhs.uk.

**Open access** This is an open access article distributed in accordance with the Creative Commons Attribution 4.0 Unported (CC BY 4.0) license, which permits others to copy, redistribute, remix, transform and build upon this work for any purpose, provided the original work is properly cited, a link to the licence is given, and indication of whether changes were made. See: https://creativecommons.org/licenses/by/4.0/.

**ORCID iDs**

Jilles M Fermont http://orcid.org/0000-0001-5042-5785
Marie Fisk http://orcid.org/0000-0002-1292-7642
Charlotte E Bolton http://orcid.org/0000-0002-9578-2249
Joseph Cheriyan http://orcid.org/0000-0001-6921-1592
Kaisa M Mäki-Petäjä http://orcid.org/0000-0001-7312-6200
Ali B Al-Hadithi http://orcid.org/0000-0003-0417-9653
Ruth Tal-Singer http://orcid.org/0000-0002-5275-8062
Michael I Polkey http://orcid.org/0000-0003-1243-8571
Angela M Wood http://orcid.org/0000-0002-7937-304X

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
