## [Reviewer comments · BMJ Open]

ARTICLE DETAILS

TITLE (PROVISIONAL)	Cardiovascular risk prediction using physical performance measures in COPD: results from a multi-centre observational study.
AUTHORS	Fermont, Jilles; Fisk, Marie; Bolton, Charlotte; MacNee, William; Cockcroft, John; Fuld, Jonathan; Cheriyan, Joseph; Mohan, Divya; Mäki-Petäjä, Kaisa; Al-Hadithi, Ali; Tal-Singer, Ruth; Müllerova, Hana; Polkey, Mike; Wood, Angela; McEniery, Carmel M.; Wilkinson, Ian

VERSION 1 – REVIEW

REVIEWER	Michael Stickland University of Alberta, Edmonton Canada
REVIEW RETURNED	30-Mar-2020

GENERAL COMMENTS	Cardiovascular disease is a known co-morbidity in patients with COPD. The purpose of this paper was to examine the value of Framingham general CV risk score, as well as additional CV markers (including aortic PWV, carotid intima media thickness, CRP, BODE, 6MW) in predicting CV events in a large cohort (n=714) of COPD patients. Framingham score was found to be a good predictor of CV hospitalizations, while novel markers such as aortic PWV and carotid intima media thickness were not. Adding BMI, 4MGS, 6MWT and BODE collectively to the Framingham risk model improved discrimination, with 6MWT primarily accounting for the improved discriminative ability of the model The rationale for the study is well presented, the methods well detailed, and overall this is a nice contribution. I do have one significant concern: The paper only reports CV-related hospitalizations (i.e. hospitalizations with ICD10 coding of CV event within the first two positions). Presumably there were COPD hospitalizations, but these are not reported. It is possible in some of the hospitalizations that COPD could be in the first position, and a CV event listed in the second position (or vice-versa). Further, in patients with advanced COPD and CV disease, it is often difficult to determine whether the hospitalization is due to CHF or COPD. This is often the case clinically, as well as with ICD10 coding. In these cases, would Framingham not also predict hospitalizations? To me, the lack of reporting & analysis of COPD hospitalizations, and the potential cross-contamination of COPD/CV hospitalizations is a weakness. Minor: What is a bit unclear to me is the term 'first event of CV hospitalisation' used by the authors. Presumably some of the patients would have had a previous CV hospitalisation prior to study enrollment. The authors should clarify that the 'first' or 'new'
---

	hospitalization refers to the first hospitalization within the current study period. The captions for Figures 1-3 state the discriminative ability for CV disease; however, I would suggest 'CV hospitalizations' is more appropriate. The comprehensiveness of the online supplement is appreciate. Nice to see the CV variables reported by site (Figures 3-5 of supplement).
--	--

REVIEWER	Dr Ann Morgan National Heart and Lung Institute, Imperial College London, UK
REVIEW RETURNED	10-Apr-2020

GENERAL COMMENTS	Comments and suggestions for authors Summary In this study, the authors assess the ability of a Framingham model to predict cardiovascular (CV) risk in a population of patients with moderate-to-severe chronic obstructive pulmonary disease (COPD) (n=714; ERICA study participants). The median age of the study cohort was 67 years; just over 60% were male, and at least a third of participants had one or more cardiovascular risk factors (hypertension/dyslipidanaemia/diabetes) at baseline. During the period of follow up (median 4.5 years), 237 study participants experienced the outcome, that is a CV-hospitalisation or CV-death. Based on the C-statistic, the authors judged that the “conventional” Framingham CV model performed moderately well in this COPD cohort (c-statistic=0.696). The model’s predictive capability (i.e. discrimination) was improved by the addition of measures of physical activity, in particular the 6MWT (0.727 v 0.696). On the basis of these findings, the authors conclude that physical inactivity is likely an important predictor of future CV disease risk and that assessment of physical performance may be a useful addition to CV risk evaluation in this population group. General comments The rationale for the reported study is sound, and represents a useful addition to the ongoing debate about the usefulness or otherwise of the screening of patients with COPD for concomitant cardiovascular disease. It would have been useful to see a little more information about the study population (the multi-centre ERICA cohort) in order to be able to assess the likelihood and potential impacts of selection bias and the generalisability of the study findings to the wider COPD population. One area of concern is the exclusion of younger patients with only mild COPD disease in whom the relative risks for CV events are known to be high (relative to the non-COPD population). However based on the information presented in Table 1, the characteristics of the study cohort at baseline (presumably on enrolment?) appear to be broadly in line with expectations for a COPD population (e.g. in terms of prevalence of CVD risk factors, gender balance), and the authors do themselves acknowledge that generalisability may indeed be limited. The authors briefly mention the QRisk score but there is no
--

explanation as to why the analysis is based on the Framingham model and not the QRisk. It is my understanding that the Framingham model/score is less reliable in the UK population and that in this respect the Qrisk performs better. Moreover I believe the Qrisk score is more widely used in UK clinical practice. However, the variables included in Qrisk2 are broadly similar to the “traditional” FRFs?

The authors should perhaps specify more clearly which version of the Framingham risk score model they are using as the basis of their analysis (merely described as the “conventional” model) as there have been several iterations of this model over time. I think the original dates from 2002 and includes age, sex, LDL cholesterol, HDL cholesterol, blood pressure (and also whether the patient is treated or not for his/her hypertension), diabetes, and smoking and estimates the 10-year risk for coronary heart disease (CHD), where CHD is defined as MI, stroke, angina and heart failure, but not diseases of the arteries such as PAD, AAA ?? The revised version, 2008, I believe excludes diabetes as a predictor but includes a wider range of CV endpoints, so including the PAD, AAA etc - that is to say the CV conditions which are included in the definition of the outcome in the present analysis. Presumably the analysis should be aiming to “match” the predictors in the model to the outcomes, and from the information provided it is not entirely transparent that this is indeed the case.

While for the most part the manuscript is well written and presented, there are instances – in the Methods especially – of ambiguity and where the sentence structure/grammar might benefit from some additional attention to improve clarity and aid reader understanding. Examples are highlighted in the specific comments below. Throughout more care needs to be taken to distinguish CV risk factors and established CV disease. Some of the references cited are a little old and some newer research has not been mentioned.

Specific comments

Introduction

2nd paragraph: people with COPD may also be at increased risk for CVD, beyond the simple presence of shared risk factors. Some researchers have suggested that a COPD diagnosis represents an additional, causal risk factor for CVD, which is not necessarily limited to acute events (MI and stroke).

Methods

Study population: You make no comment as to how patients were enrolled/recruited into the study nor indeed when the study began. Presumably baseline patient characteristics and clinical measures were assessed at this point? Exclusion/inclusion of patients on the basis of their CVD/exacerbation history is not clear; did you exclude patients who had previously been hospitalised for a CV event (this is what is meant by “first” occurrence or is this just first occurrence in the follow up period)? Were patients required to be exacerbation free for a period of at least four weeks prior to their baseline assessment?

Clinical measures: presumably smoking status was self-reported, and BMI, BP and cholesterol tests were also conducted at baseline.

Outcomes (CV-hospitalisation and CV-death): The manner of the patient data linkage (ERICA cohort data with HES/ONS data) is not clear (wording could be improved). Also, the way in which CV events were identified from HES hospital records is not well explained, nor justified (in terms of any assumptions made). Were events identified on the basis of the presence of a relevant ICD 10 code in the first or second position (i.e. primary and second diagnosis) in the first finished consultant episode (FCE) or any (FCE) that collectively make up a spell in hospital? Some researchers opt not to use a diagnosis in the second position to identify incident (first) events) as sometimes diagnoses recorded in the second and subsequent positions in each successive FCE denote comorbidities. The authors may like to consider conducting a sensitivity analysis to explore the robustness of their results to various assumptions made when developing their strategies for identifying CV events in HES data. I note that it is reported in the results section that a significant proportion of events stem from CV ICD-10 codes recorded as a secondary diagnosis – which might suggest a prevalent or underlying condition as opposed to a CV being the primary reason for hospitalisation. Given the relatively high number of outcome events recorded in this cohort, I would suggest that some form of sensitivity analysis along these lines is warranted.

Predictors: Presumably only those with a smoking history were included (never smokers were excluded as not meeting the criteria for COPD?), so more accurately participants divide into current and ex-smokers (as opposed to current and non-smokers)?

Statistical analysis: The description of the statistical analysis is rather brief, and limits the readers' ability to judge appropriateness of the analysis strategy employed. While I accept there is the inevitable challenge of keeping to a strict word limit, a little more detail could be provided, if not in the main text but in the Supplementary material (which is limited to a discussion of the methods employed to handle missing data). For instance, it is not made clear why the alternative measures (i.e. non-traditional FRFs) were adjusted for age and sex, and stratified by study centre. I'm guessing the latter, stratification by study centre, might be related to the need to account for the clustered nature of the data set (this being a multi-centre study). In addition, is there a possible issue with competing risks (in the sense that some participants may have died from a non-CV cause without having had the opportunity to experience the (non-fatal) outcome; it is noted in the results that only 13% of deaths are from a CV-cause)? Again sensitivity analyses might usefully explore the impact of missing data on study findings, although the level of missingness does not seem to be particularly high.

Results

Note: Supplementary table numbering needs double checking.

Descriptive analysis: it might be helpful to also include the median Framingham risk score (which could be calculated at baseline from the data collected?) and possibly report what percentage of participants whose score was over say 20% were already receiving primary prevention (antihypertensives, statins) to manage their cardiovascular risk factors. This is of interest as it is believed that a

	not insubstantial proportion of COPD patients are undertreated for CV disease risks. Table 1: The column heading is a misleading [No of participants (%), unless otherwise indicated] Traditional/additional risk factors: Figures 1-3: an additional comment to explain how readers should interpret the c-index forest plot might aid readability/user-friendliness. This could be incorporated into the figure legend? Mortality: Break down by age might also be insightful: older COPD patients tend to die of respiratory causes; the younger ones are the more likely to die of a cardiovascular cause. Discussion Paras 2/3, 5. The discussion of the performance of the Framingham model in other populations needs more careful qualification. For instance, Ref 6 implies that it does not perform particularly well in older populations: how does the age of that population compare to this one? The comparison with ref 10 needs to distinguish established CVD and CVD risk factors. Perhaps given the limitations in making direct comparisons with other studies which calculate discrimination, perhaps more could be said about the lack of predictive role of some of the other additional measures (e.g. aPWV, inflammatory markers) in COPD which have been shown to improve CV risk prediction on other, ie non-COPD, populations? Perhaps provide some suggestions as to why this appears to be the case in the COPD patient population? Para 6. The negative association between SBP and the outcome is only of borderline significance, especially in the fully adjusted model, and perhaps not too much can be read in to this finding. That said, the comments made do seem plausible, but this is a tricky area to unpick given the possibility of confounding by indication. Paras 7/8: I have previously commented on the apparently high number of subjects who experienced the outcome, and note that this issue was raised as a limitation. However I would re-iterate that a sensitivity analysis to explore the effect of using events recorded in the second position would strengthen confidence in the reported results. Presumably people who had a prior history of hospitalisation for MI etc were excluded but those with a history of CV risk factors (hypertension, atrial fibrillation etc) were not excluded? So when you refer to “pre-existing CV morbidity at baseline” you mean risk factors not established disease? Point needs clarifying (see also comments under Methods). Cohort age may also have something to do with the relatively small number of CV deaths in this group. In most COPD populations, deaths are roughly equally divided between respiratory, cancer and CVD, although as previously mentioned younger patients are proportionately more likely to die from a CVD cause than older patients. Also there is recent evidence to suggest that the proportion of CVD deaths has begun to decline in COPD (see Gayle A et al, 2019), possibly for the reasons you suggest. Para 9/10 (limitations): Is it possible to comment on what the effect of having a shorter period of follow up (less than 10 years) is likely to be? Would this likely adversely impact on predictive ability?
--	--

REVIEWER	Lisa Lix University of Manitoba Winnipeg, MB
REVIEW RETURNED	15-May-2020

GENERAL COMMENTS	This is a well-written manuscript and the methodology is clear. However, a few points of clarification are needed on the methodology: (1) In the article summary (strengths and limitations of this study) the authors note “a robust multivariable prediction model...” was used. What is meant by the word “robust”? Is this term necessary? Also, the authors note that “Generalisability of our results is limited to those with mainly moderate COPD...” Can the phrase “mainly moderate” be explained? (2) How was the proportional hazard assumption assessed for the multivariable Cox models? This information should be added to the statistical analysis section. (3) The authors note that the analyses were stratified by study site? Can they please explain in more detail how this stratification was undertaken within the modeling (i.e., in the Stata or R code)? This information could be added to the supplemental information on the statistical analysis. (4) In the Results section and Supplementary Figure 6, please provide the percentage of the original number screened who were excluded (not just the frequencies). (5) How did the authors determine that they had captured a first event of CV hospitalization? Did they have information for the cohort prior to the baseline measurement occasion to determine that there had been no CV hospitalizations prior to the baseline measurement ? (6) While the authors have reported a measure of discriminative performance for their models, other measures of model performance would be beneficial for the reader. In particular, please provide information about model calibration, which describes how accurately the estimates or predictions of survival from the model reflect survival in the observed data. (7) Were there any issues with collinearity amongst the variables included in the Cox models? (8) In the Discussion section, please provide a published reference for the use of deprivation scores to predict CV risk. Not all readers may understand how deprivation is measured, or what the concept means.
---

VERSION 1 – AUTHOR RESPONSE

Reviewer: 1

Cardiovascular disease is a known co-morbidity in patients with COPD. The purpose of this paper was to examine the value of Framingham general CV risk score, as well as additional CV markers (including aortic PWV, carotid intima media thickness, CRP, BODE, 6MWT) in predicting CV events in a large cohort (n=714) of COPD patients. Framingham score was found to be a good predictor of CV hospitalizations, while novel markers such as aortic PWV and carotid intima media thickness were not. Adding BMI, 4MGS, 6MWT and BODE collectively to the Framingham risk model improved discrimination, with 6MWT primarily accounting for the improved discriminative ability of the model

The rationale for the study is well presented, the methods well detailed, and overall this is a nice contribution. I do have one significant concern:

The paper only reports CV-related hospitalizations (i.e. hospitalizations with ICD10 coding of CV event within the first two positions). Presumably there were COPD hospitalizations, but these are not reported. It is possible in some of the hospitalizations that COPD could be in the first position, and a

CV event listed in the second position (or vice-versa). Further, in patients with advanced COPD and CV disease, it is often difficult to determine whether the hospitalization is due to CHF or COPD. This is often the case clinically, as well as with ICD10 coding. In these cases, would Framingham not also predict hospitalizations? To me, the lack of reporting & analysis of COPD hospitalizations, and the potential cross-contamination of COPD/CV hospitalizations is a weakness.

- A. Patients had multiple hospital visits (with primary and secondary ICD-10 coding) during the study period. Some patients were admitted to hospital for acute exacerbation of COPD (AECOPD) [reported by either primary or secondary position ICD-10 position], were admitted for other causes, or had both COPD and CV admissions during the study period. In our cohort COPD hospitalisations were primarily recorded in the primary position, and CV events in the secondary. When selecting the CV events we gave priority to a CV event in the primary position, which was the case for 43 cases. We then selected CV events in the secondary position, if not already reported in the primary position, which was the case for 189 individuals. We include these numbers in the Results section. In addition, we have included a table in the supplementary file indicating the number of individuals with a CV event reported in the primary or secondary position, and if they had a recorded COPD hospitalisation during the study period (see supplementary table 2). Please note that in this manuscript we exclusively focussed on CV admission.

We added the following text to the Methods section in the supplementary file: “Data were cleaned for episode status, and inpatient (i.e. hospitalised) CV episodes were identified based on classifications used by the Emerging Risk Factors Collaboration¹. Cardiovascular events were extracted from both primary and secondary positions of ICD-10. Priority was given to CV events reported in the primary position (n = 43). Following this, we selected CV events in the secondary position (n = 189), which had no recording of a CV event in the primary position. Only episodes during the study follow-up were evaluated.”

- A. In addition, hospital admissions due to acute exacerbations of COPD (AECOPD) have been reported previously by us (Fermont *et al.* doi.org/10.1371/journal.pone.0228940). For AECOPD there is an algorithm available using validated criteria to extract events from both primary and secondary position and classifies these as definite, potential or possible AECOPD (Rothnie *et al.* doi.org/10.2147/CLEP.S117867). Note that this validated algorithm extracts events from both primary and secondary positions. A similar algorithm to identify CV events using ICD-10 coding does not exist unfortunately. There is considerable variation in the published literature how CV events have been defined and identified. We did consider extracting CV events from the primary position only. However, there were not enough events for any meaningful analysis and have therefore decided to include both positions. Previously CV disease endpoints have been validated using the Clinical Practice Research Datalink (CPRD) classification (CALIBER EHR phenotyping) algorithm (Herrett *et al.* [doi: 10.1111/j.1365-2125.2009.03537.x](https://doi.org/10.1111/j.1365-2125.2009.03537.x)) combined with extensive clinical input (Bell *et al.* doi.org/10.1136/bmj.j909). We did not, however, have access to CPRD data and instead defined CV disease based on classifications used by the Emerging Risk Factors Collaboration (Emerging Risk Factors Collaboration [doi: 10.1007/s10654-007-9165-7](https://doi.org/10.1007/s10654-007-9165-7)).

We have added the following text to the Methods section: “We defined the primary outcome as first reported occurrence (since study enrolment) of fatal or non-fatal hospitalised CVD, where CVD was defined as diseases of the arteries, stroke or heart failure (see supplementary table 1) based on classifications used by the Emerging Risk Factors Collaboration.[27]” We do describe this as a limitation in the Discussion.

- A. There is indeed potential cross-contamination of COPD/CV hospitalizations. Often there is misclassification of morbidity and mortality when data are obtained from routine sources such as death certificates (Macnee *et al.* doi.org/10.1513/pats.200807-071TH). We acknowledge, however, this as a limitation and have added the following text to the limitation section in the Discussion: “It is possible our models predict non-CV hospitalisation due to, for example, potential cross-contamination of COPD and CV hospitalisation as a result of misclassification of morbidity and mortality that often occurs when data are obtained from routine sources.[43]”

Minor:

What is a bit unclear to me is the term ‘first event of CV hospitalisation’ used by the authors. Presumably some of the patients would have had a previous CV hospitalisation prior to study enrolment. The authors should clarify that the ‘first’ or ‘new’ hospitalization refers to the first hospitalization within the current study period.

- A. The study period covered the time from study enrolment until the end of study, or death. Some individuals, however, may have been admitted to hospital shortly before study enrolment and these events will have not been captured electronically and made available to us..

We have updated the text in the methods section as follows: “The aims of our study, defined by new occurrence (first event since study enrolment) of fatal or non-fatal ... COPD.

The captions for Figures 1-3 state the discriminative ability for CV disease; however, I would suggest ‘CV hospitalizations’ is more appropriate.

- A. We have updated the figure captions and replaced the text cardiovascular hospitalisations with “fatal or non-fatal hospitalised cardiovascular disease.”

The comprehensiveness of the online supplement is appreciate. Nice to see the CV variables reported by site (Figures 3-5 of supplement).

Reviewer: 2

Summary

In this study, the authors assess the ability of a Framingham model to predict cardiovascular (CV) risk in a population of patients with moderate-to-severe chronic obstructive pulmonary disease (COPD) (n=714; ERICA study participants). The median age of the study cohort was 67 years; just over 60% were male, and at least a third of participants had one or more cardiovascular risk factors (hypertension/dyslipidaemia/diabetes) at baseline. During the period of follow up (median 4.5 years), 237 study participants experienced the outcome, that is a CV-hospitalisation or CV-death.

Based on the C-statistic, the authors judged that the “conventional” Framingham CV model performed moderately well in this COPD cohort (c-statistic=0.696). The model’s predictive capability (i.e. discrimination) was improved by the addition of measures of physical activity, in particular the 6MWT (0.727 v 0.696). On the basis of these findings, the authors conclude that physical inactivity is likely an important predictor of future CV disease risk and that assessment of physical performance may be a useful addition to CV risk evaluation in this population group.

General comments

The rationale for the reported study is sound, and represents a useful addition to the ongoing debate about the usefulness or otherwise of the screening of patients with COPD for concomitant cardiovascular disease. It would have been useful to see a little more information about the study population (the multi-centre ERICA cohort) in order to be able to assess the likelihood and potential impacts of selection bias and the generalisability of the study findings to the wider COPD population.

- A. We have added the following text to the Methods section: “Participants had a clinical diagnosis of COPD, smoking history of at least ten pack years, FEV₁ / FVC ratio <0.7 and FEV₁ ≤80% of predicted normal lung function, and were aged >40 years old and clinically stable for >4 weeks.” We do, however, reference to the study protocol with full details on the ERICA cohort.

One area of concern is the exclusion of younger patients with only mild COPD disease in whom the relative risks for CV events are known to be high (relative to the non-COPD population). However based on the information presented in Table 1, the characteristics of the study cohort at baseline (presumably on enrolment?) appear to be broadly in line with expectations for a COPD population (e.g. in terms of prevalence of CVD risk factors, gender balance), and the authors do themselves acknowledge that generalisability may indeed be limited.

- A. We are aware of the publication by Morgan et al (2018) where the authors emphasize the importance of CV risk screening in middle-aged individuals. However, the usual onset of COPD is around the age of 40, matching the selection criteria of our study, and with a median age of 67 years our cohort does reflect a typical COPD population. These data were collected at enrolment. We kindly point out the publication by Mohan *et al.* doi.org/10.3109/15412555.2014.898031 who further describe details on the selection criteria of our cohort. As the reviewer implies, however, our methodology might have overlooked patients with undiagnosed COPD who can nevertheless experience CV events.

The authors briefly mention the QRisk score but there is no explanation as to why the analysis is based on the Framingham model and not the QRisk. It is my understanding that the Framingham model/score is less reliable in the UK population and that in this respect the QRisk performs better. Moreover I believe the QRisk score is more widely used in UK clinical practice. However, the variables included in QRisk2 are broadly similar to the “traditional” FRFs?

- A. The QRISK would indeed have been more appropriate for our UK population. We did consider using the QRISK. However, we did not have all the variables required to estimate the QRISK. For example, we did not have baseline data on the presence of chronic kidney disease, atrial fibrillation, and rheumatoid arthritis. In addition, we did not have details on deprivation. We have decided to remove the term Framingham and are now referring to conventional CVD risk factors as we feel this is more appropriate.

The risk factors of interest section in the Methods now reads: “Conventional CVD risk factors included age, sex, self-reported smoking status [current/ex-smoker], HDL cholesterol, total cholesterol, systolic blood pressure (SBP), diabetes [yes/no] and treatment for high blood pressure [yes/no].”

The authors should perhaps specify more clearly which version of the Framingham risk score model they are using as the basis of their analysis (merely described as the “conventional” model) as there have been several iterations of this model over time. I think the original dates from 2002 and includes age, sex, LDL cholesterol, HDL cholesterol, blood pressure (and also whether the patient is treated or not for his/her hypertension), diabetes, and smoking and estimates the 10-year risk for coronary heart disease (CHD), where CHD is defined as MI, stroke, angina and heart failure, but not diseases of the arteries such as PAD, AAA ?? The revised version, 2008, I believe excludes diabetes as a predictor but includes a wider range of CV endpoints, so including the PAD, AAA etc - that is to say the CV conditions which are included in the definition of the outcome in the present analysis. Presumably the analysis should be aiming to “match” the predictors in the model to the outcomes, and from the information provided it is not entirely transparent that this is indeed the case.

- A. Please see previous point on Framingham/QRISK. We have removed the term Framingham and are now referring to conventional CVD risk factors. The aim of our study, however, was not to evaluate Framingham both because this has been done before but also because our cohort was not big enough for that. Rather we sought to understand whether newer measures such as aPWV or CIMT might improve when added to conventional CVD risk factors. Unexpectedly we found that these did not improve the risk assessment, although a measure of exercise capacity did.

The events, as described in supplementary Table 1, include MI, stroke, angina, heart failure, and other diseases including PAD and AAA. CV disease was based on classifications used by the Emerging Risk Factors Collaboration, and are hard CV outcomes.

We have provided additional details on the type of events captured as a footnote in the Results section and included the following text: "Peripheral arterial disease (n = 77), diseases of arteries, arterioles and capillaries (n = 6), angina (n = 24), unstable angina (n = 2), coronary heart disease not otherwise specified (n = 58), acute myocardial infarction (MI), and certain current complications following acute MI (n = 8), cerebral infarction (n = 9), stroke, not specified as haemorrhage or infarction (n = 1), other stroke (n = 14), heart failure (n = 28), abdominal aortic aneurysm (n = 5)."

While for the most part the manuscript is well written and presented, there are instances – in the Methods especially – of ambiguity and where the sentence structure/grammar might benefit from some additional attention to improve clarity and aid reader understanding. Examples are highlighted in the specific comments below. Throughout more care needs to be taken to distinguish CV risk factors and established CV disease. Some of the references cited are a little old and some newer research has not been mentioned.

- A. We have changed the text in the Cardiovascular hospitalisation and mortality section of the Methods as follows: "Cardiovascular hospitalisation and mortality data were extracted from the linked hospital admission data and death certificates. Non-fatal CV episodes were extracted from both primary and secondary international statistical classification of diseases and related health problems 10th revision coding (ICD-10) positions. Causes of death were adjudicated by CV and pulmonary physicians. We defined the primary outcome as first reported occurrence (since study enrolment) of fatal or non-fatal hospitalised CVD, where CVD was defined as diseases of the arteries, stroke or heart failure (see supplementary table 1) based on classifications used by the Emerging Risk Factors Collaboration.[27] Time to primary outcome was calculated from the difference between the baseline visit date (starting December 2011) and either the date of death or first hospitalised CV attendance up to November 2017, when follow-up discontinued. Secondary outcomes of interest were all-cause and cause-specific mortality (defined as CV, pulmonary, cancer, or other)."
- A. We thank the reviewer for pointing this out. We have removed the Kannel et al. reference dating from 1979 and included the following reference: Mora et al. (2005) doi.org/10.1161/CIRCULATIONAHA.105.542993. In addition, we have added the following text and reference in the Discussion: "Especially those aged under 65 years may benefit most from active CVD assessment, according to Morgan et al.[36]" Morgan et al. doi.org/10.1177/1753465817750524.

Despite being older references, we do think references [22 Wilkinson et al. [doi:10.1097/00004872-199816121-00033](https://doi.org/10.1097/00004872-199816121-00033), 23 Csanyi and Egervari [PMID: 8795305](https://pubmed.ncbi.nlm.nih.gov/8795305/) and 25 Guralnik et

al. [doi:10.1093/geronj/49.2.m85](https://doi.org/10.1093/geronj/49.2.m85)] referenced in the Methods section are still appropriate as these are guidelines/descriptions of testing CIMT (1996), SPPB (1994), and PWV and Alx (1998). Reference 17 Hole *et al* [doi:10.1136/bmj.313.7059.711](https://doi.org/10.1136/bmj.313.7059.711) dates from 1996 but is a seminal study reporting on the relationship between lung function and mortality.

VERSION 2 – REVIEW

REVIEWER	Michael Stickland Division of Pulmonary Medicine, Department of Medicine, University of Alberta, Edmonton, AB, Canada
REVIEW RETURNED	13-Jul-2020

GENERAL COMMENTS	Paper provides an important contribution to the area. Thank you.
--

REVIEWER	Dr Ann Morgan National Heart and Lung Institute Imperial College London UK
REVIEW RETURNED	19-Jul-2020

GENERAL COMMENTS	Please note that I have previously reviewed this paper, and therefore my comments below outline the main outstanding concerns that I have, as opposed to constituting a full review of the current version of the manuscript. While the majority of the concerns raised by myself and the other reviewers have been adequately addressed in the revised version, there are a couple of issues that I feel warrant further clarification. My main outstanding concern lies with the definition of the main outcome - which is hospitalisation for a cardiovascular "episode" or CV death (MI, stroke, AAA, PAD, heart failure, angina). I accept that basing the primary analysis on CVD recorded as either a primary or secondary diagnosis may be an acceptable approach, given your sample size. However if my interpretation of Table E2 is correct, you have also included in the tally of patients who have experienced the outcome, those with an E11.9 ICD-10 code as their secondary diagnosis; a proportion of these patients were admitted for an AECOPD (ie this was recorded as the primary diagnosis)? I may be wrong but I think ICD-10 code E11.9 is diabetes (a common comorbidity in COPD). If it is, I am not entirely convinced that it is reasonable to include these patients (around 65) among the 237 listed as having had a CVD hospitalisation? I'm not sure whether the exclusion of these patients would compromise the power of the study, but given this possibility and the additional limitations you set out, I am left wondering whether the analysis really supports the rather strongly worded conclusion - namely that adding physical performance measures "significantly improved the predictive discrimination of a CV risk model"? My remaining concerns are minor and relate to the standard of the written English. This could be improved in places to improve sense and clarity, especially for readers who may not be that familiar with HES data.
---

REVIEWER	Lisa Lix
-----------------	----------

	University of Manitoba, Canada
REVIEW RETURNED	16-Jul-2020

GENERAL COMMENTS	The authors have thoroughly addressed most of the comments and suggested revisions posed by the reviewers. However, there are a few points of clarification needed. (1) The authors indicate that the Hosmer-Lemeshow test was used to evaluate goodness of fit (see page 10). However, this test was developed for a logistic regression model, and not for the Cox semi-parametric model used in this study. Adaptations of the Hosmer-Lemeshow test have been developed for the Cox model. Can the authors please appropriately reference the adaptation that was used in this study? Also, information about the outcomes of the assessment of model fit using the adaptation of the Hosmer-Lemeshow test should be added to the results section of the manuscript. (2) In the methods section, the authors note that model calibration was also assessed using the Brier score. However, they have not provided any information about how to interpret the Brier score results in the methods section (there is a brief mention in the figure legends only). While lower Brier scores are better, what are the maximum and minimum possible values and how should the scores be interpreted. Moreover, while Brier score statistics are reported in the figure legends, the authors should also discuss these (briefly) in the text of the manuscript. (3) In the Discussion section on page 19, please clarify what is meant by the sentence "We did not have spell data available." (4) There are some grammatical errors to be fixed in the new material provided in the discussion section. For example, "predictive ability would diminishes because ageing is a strong predictor...should be "predictive ability would diminish because ageing is a strong predictor". Please review the new material for any further grammatical errors that require correction.
---

VERSION 2 – AUTHOR RESPONSE

Reviewer: 1

Please leave your comments for the authors below
 Paper provides an important contribution to the area. Thank you.

Reviewer: 2

Please leave your comments for the authors below
 Please note that I have previously reviewed this paper, and therefore my comments below outline the main outstanding concerns that I have, as opposed to constituting a full review of the current version of the manuscript.

While the majority of the concerns raised by myself and the other reviewers have been adequately addressed in the revised version, there are a couple of issues that I feel warrant further clarification.

My main outstanding concern lies with the definition of the main outcome - which is hospitalisation for a cardiovascular "episode" or CV death (MI, stroke, AAA, PAD, heart failure, angina). I accept that basing the primary analysis on CVD recorded as either a primary or secondary diagnosis may be an acceptable approach, given your sample size. However if my interpretation of Table E2 is correct, you have also included in the tally of patients who have experienced the outcome, those with an E11.9 ICD-10 code as their secondary diagnosis; a proportion of these patients were admitted for an AECOPD (ie this was recorded as the primary diagnosis)? I may be wrong but I think ICD-10 code E11.9 is diabetes (a common comorbidity in COPD). If it is, I am not entirely convinced that it is reasonable to include these patients (around 65) among the 237 listed as having had a CVD hospitalisation? I'm not sure whether the exclusion of these patients would compromise the power of the study, but given this possibility and the additional limitations you set out, I am left wondering whether the analysis really supports the rather strongly worded conclusion - namely that adding physical performance measures "significantly improved the predictive discrimination of a CV risk model"?

- A. ICD-10 coding E11.9 refers to non-insulin-dependent diabetes mellitus without complications. In defining the outcome measure we based this on existing literature. However, we have removed the E10, E11, and R96 ICD-10 coding from our list, and re-ran all the analysis. The main text, tables (including supplementary) and figures have been updated accordingly.

Multiple individuals included for an event related to the above-mentioned ICD-10 coding had alternative CV events, albeit later during the study period. The total number of events has reduced from 237 to 192. Re-analysing our data has resulted in the following changes:

- Median follow-up time changed from 4.5 to 4.6 years.
- Total number of CV events is 192 including 6 cardiac deaths
- Slight changes in HRs and C-indices
- BMI, glucose and diabetes are not significantly associated with the outcome anymore, after including conventional CV risk factors
- Systolic blood pressure and Augmentation index (AIx) are now significantly associated with the outcome, when including conventional CV risk factors. However, AIx did not improve the discriminative ability of the CV risk model.
- All variables combined do improve the discriminative ability but still is primarily driven by 6MWT.

My remaining concerns are minor and relate to the standard of the written English. This could be improved in places to improve sense and clarity, especially for readers who may not be that familiar with HES data.

- A. We have included the following reference, Herbert *et al.* (doi.org/10.1093/ije/dyx015), and footnote to the study design and participants section in the Methods: "Hospital episode statistics (HES) data is a database that includes details of all hospital admissions, accident and emergency department visits and outpatient appointments at an individual patient level."

In addition, we have improved the readability of the manuscript throughout.

Reviewer: 3

Please leave your comments for the authors below

The authors have thoroughly addressed most of the comments and suggested revisions posed by the reviewers. However, there are a few points of clarification needed.

(1) The authors indicate that the Hosmer-Lemeshow test was used to evaluate goodness of fit (see page 10). However, this test was developed for a logistic regression model, and not for the Cox semi-

parametric model used in this study. Adaptations of the Hosmer-Lemeshow test have been developed for the Cox model. Can the authors please appropriately reference the adaptation that was used in this study? Also, information about the outcomes of the assessment of model fit using the adaptation of the Hosmer-Lemeshow test should be added to the results section of the manuscript.

- A. We thank the reviewer for pointing this out. The text should have read Gronnesby and Borgan test instead of Hosmer-Lemeshow. We have updated the text and figures accordingly. Goodness-of-fit test was performed using the *stcoxgof* command in STATA. “*stcoxgof*- is a post-estimation command testing the goodness of fit after a Cox model.”

We have included the following text to the Methods: “...calibration (i.e. Gronnesby and Borgan test and Brier score).

We have added the following text to the Results section: “Calibration tests indicate good model fit (figure 3).”

(2) In the methods section, the authors note that model calibration was also assessed using the Brier score. However, they have not provided any information about how to interpret the Brier score results in the methods section (there is a brief mention in the figure legends only). While lower Brier scores are better, what are the maximum and minimum possible values and how should the scores be interpreted. Moreover, while Brier score statistics are reported in the figure legends, the authors should also discuss these (briefly) in the text of the manuscript.

- A. There are no defined cut off points for the Brier score. Instead, when using the same outcome data the Brier score allows for comparing the performance of a model with a reference model.

We have added the following text to the Methods section: “Brier scores range from 0-1 (0 is perfect accuracy and 1 perfect inaccuracy), and allows to compare performance of a model with a reference model.”

We have added the following text to the Results section: “The model including 6MW had a better Brier score relative to the CV risk model (0.123 vs. 0.129, respectively).”

(3) In the Discussion section on page 19, please clarify what is meant by the sentence "We did not have spell data available."

- A. In the first review it was asked if events were identified through the first finished consultant episode (FCE) or any (FCE) that collectively make up a spell in hospital. Therefore we added the text “We did not have spell data available” in the limitation section of the Discussion. However, we have decided to remove this line because it does actually not affect our analysis or outcomes and might confuse the reader.

There is a difference between a spell (patient's entire stay in hospital) and FCE (the period of time a patient spends under the care and responsibility of one consultant team). Multiple episodes may occur during one spell. For example, some patients with an initial diagnosis may go on to have a subsequent event within a spell. However, we used episode in identifying first event since study enrolment and this will therefore not impact our outcome. It would, for example, be relevant when evaluating hospital payment/financing ([doi: 10.1136/bmj.329.7476.1207](https://doi.org/10.1136/bmj.329.7476.1207)).

(4) There are some grammatical errors to be fixed in the new material provided in the discussion section. For example, "predictive ability would diminishes because ageing is a strong predictor...should be "predictive ability would diminish because ageing is a strong predictor". Please review the new material for any further grammatical errors that require correction.

- A. We have checked the revisions for errors and updated the text as pointed out by the reviewer, as follows: "For example, a too short time period may result in an insufficient number of events, whilst over a longer time period, e.g. 20 years, the predictive ability would diminish because ageing is a strong predictor."

In addition, please see our response to the last comment of Reviewer 2.

VERSION 3 – REVIEW

REVIEWER	Ann Morgan National Heart and Lung Institute Imperial College London London, UK
REVIEW RETURNED	27-Oct-2020

GENERAL COMMENTS	Given that the authors have re-done the analyses but excluding the diabetes codes, and have improved the language, I happy to recommend this paper for publication.
---

REVIEWER	Lisa Lix University of Manitoba, Canada
REVIEW RETURNED	30-Aug-2020

GENERAL COMMENTS	The authors have nicely responded to the feedback provided by the reviewers. The manuscript is strengthened. However, there are some minor revisions that require attention. In the Methods section, the authors note that they performed the Gronnesby and Borgan test to assess model calibration. In the figures they report chi-square statistics associated with this test. It would be helpful for readers to have information about the Gronnesby and Borgan test on pages 9-10 of the manuscript. What exactly is being tested? What test statistic is used? The authors have provided information about the Brier score and the c-statistic on these pages, but not provided an equivalent level of information about the Gronnesby and Borgan test. As well, when chi-squared goodness of fit test statistics are provided in the figure legends, the authors must be explicit in noting that these are for the Gronnesby and Borgan test. Finally, chi-squared test statistics for model fit are provided in the legends for both figures 2 and 3, but only mentioned in the text of the manuscript for figure 3. For figure 2, please provide more information in the text of the text of the manuscript about the results of the model calibration assessment. Second, in each of figures 1-3, there is a change required in the right-most panel (i.e., where the change in the value of the c-statistic is reported). The phrase "C-index change" should be renamed as "C-statistic change" As well, the 95% confidence intervals should be
---

	added for each change statistic estimate, as is already indicated in the horizontal axis label; only the point estimate is currently provided in the figures.
--	---

VERSION 3 – AUTHOR RESPONSE

Reviewer: 3

Comments to the Author

The authors have nicely responded to the feedback provided by the reviewers. The manuscript is strengthened. However, there are some minor revisions that require attention.

- Q. In the Methods section, the authors note that they performed the Gronnesby and Borgan test to assess model calibration. In the figures they report chi-square statistics associated with this test. It would be helpful for readers to have information about the Gronnesby and Borgan test on pages 9-10 of the manuscript. What exactly is being tested? What test statistic is used? The authors have provided information about the Brier score and the c-statistic on these pages, but not provided an equivalent level of information about the Gronnesby and Borgan test. As well, when chi-squared goodness of fit test statistics are provided in the figure legends, the authors must be explicit in noting that these are for the Gronnesby and Borgan test. Finally, chi-squared test statistics for model fit are provided in the legends for both figures 2 and 3, but only mentioned in the text of the manuscript for figure 3. For figure 2, please provide more information in the text of the text of the manuscript about the results of the model calibration assessment.
- A. Gronnesby and Borgan (GB) test, a post-estimation command testing for the goodness of fit after a Cox model, divide the observations into groups based on their estimated risk score. The GB test (a calibration test for Cox models) is similar to the Hosmer-Lemeshow test since the martingale residuals correspond to the observed minus expected number of events for each subject.
- The following text has been added to the statistical analysis section of the Methods: Gronnesby and Borgan is an overall calibration test for Cox models based on grouping individuals by their estimated risk score and compares observed and model-based expected events within each group (a P-value > α suggests no difference).
- Legends of figures 2 and 3 have been updated and now read: "Gronnesby and Borgan goodness of fit..."
- In addition, the following text has been added to the Results section: "Calibration tests indicate good model fit (figures 2 and 3)."
- Q. Second, in each of figures 1-3, there is a change required in the right-most panel (i.e., where the change in the value of the c-statistic is reported). The phrase "C-index change" should be renamed as "C-statistic change" As well, the 95% confidence intervals should be added for each change statistic estimate, as is already indicated in the horizontal axis label; only the point estimate is currently provided in the figures.
- A. Figures have been updated per suggestion of the reviewer. Please note that the CIs are already included but merely not visible due to the intervals being too small.

Reviewer: 2

Comments to the Author

Given that the authors have re-done the analyses but excluding the diabetes codes, and have improved the language, I happy to recommend this paper for publication.